# H-InDex: Visual Reinforcement Learning with Hand-Informed Representations for Dexterous Manipulation

**Yanjie Ze**[12] **Yuyao Liu**[3*] **Ruizhe Shi**[3*] **Jiaxin Qin**[4]
**Zhecheng Yuan**[31] **Jiashun Wang**[5] **Huazhe Xu**[316]

[1]Shanghai Qi Zhi Institute  [2]Shanghai Jiao Tong University  [3]Tsinghua University, IIIS
[4]Renmin University of China  [5]Carnegie Mellon University  [6]Shanghai AI Lab
[*]Equal contribution. Order is decided by coin flip.

**yanjieze.com/H-InDex**

## Abstract

Human hands possess remarkable dexterity and have long served as a source of inspiration for robotic manipulation. In this work, we propose a human **Hand-Informed** visual representation learning framework to solve difficult **Dex**terous manipulation tasks (**H-InDex**) with reinforcement learning. Our framework consists of three stages: *(i)* pre-training representations with 3D human hand pose estimation, *(ii)* offline adapting representations with self-supervised keypoint detection, and *(iii)* reinforcement learning with exponential moving average BatchNorm. The last two stages only modify $0.36\%$ parameters of the pre-trained representation in total, ensuring the knowledge from pre-training is maintained to the full extent. We empirically study **12** challenging dexterous manipulation tasks and find that **H-InDex** largely surpasses strong baseline methods and the recent visual foundation models for motor control. Code is available at **yanjieze.com/H-InDex**.

## 1 Introduction

Humans can adeptly tackle intricate and novel dexterous manipulation tasks. However, multi-fingered robotic hands still struggle to achieve such dexterity efficiently. Recent progress in representation learning for visuomotor tasks has proved that pre-trained universal representations may accelerate robot learning manipulation tasks [17, 18, 26, 30]. In light of previous success, the similar morphology between human hands and robotic hands begs the question: can robotic hands leverage representations learned from human hands for achieving dexterity?

In this paper, we propose **Hand-In**formed visual reinforcement learning framework for **Dex**terous manipulation (**H-InDex**) that uses

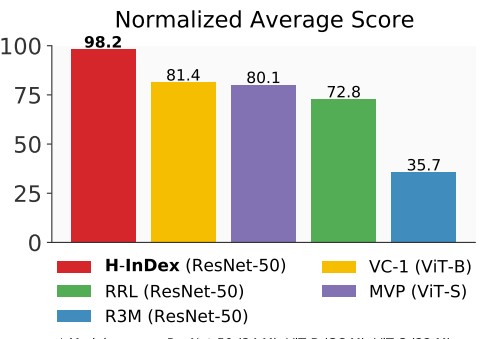

Figure 1: **Normalized average score** for our algorithm H-InDex and the baselines (VC-1 [17], MVP [30], R3M [18], and RRL [26]).

and adapts visual representations from human hands to boost robotic hand dexterity. Our framework consists of three stages:

- *Stage 1: Pre-training representations* with 3D human hand pose estimation, where we adopt the feature encoder from an off-the-shelf 3D hand pose estimator FrankMocap [24].
- *Stage 2: Offline adapting representations* with self-supervised keypoint detection, where we freeze the convolutional layers in the pre-trained representation and only finetune the affine

37th Conference on Neural Information Processing Systems (NeurIPS 2023), New Orleans.

transformations in BatchNorm layers (**0.18**% parameters of the entire model). Such minimal modification of the pre-trained representations ensures that human dexterity is retained to the maximum extent and adapts the human hand representations into the target robotic domain.

- *Stage 3: Reinforcement learning* with exponential moving average (EMA) BatchNorm and the adapted representations. EMA operates to dynamically update the mean and variance in BatchNorm layers, to further make the model adapt to the progressive learning stages.

In contrast to previous works that also learn representations from human videos [17, 18, 30], there are two major benefits of our framework: *i)* H-InDex explicitly learn human dexterity by forcing the model to predict the 3D hand pose instead of predicting or discriminating pixels unsupervisedly using masked auto-encoding [9] or time contrastive learning [25]; *ii)* H-InDex directly adopts the off-the-shelf visual model that is designed to capture human hands rather than training large models on large-scale datasets for specific robotic tasks. These two points combined demonstrate a new cost-effective way to solve robotic tasks such as dexterous manipulation by leveraging existing visual models that are originally and only designed for human understanding.

To show the effectiveness of H-InDex, we experiment on **12** challenging visual dexterous manipulation tasks from Adroit [23] and DexMV [22]. We mainly report episode returns instead of success rates to better show how well the robots solve the tasks. In comparison with several strong visual foundation models for motor control, H-InDex largely surpasses all of them as shown in Figure 1.

To summarize, our contributions are three-fold:

- We propose a novel visual reinforcement learning framework called **H-InDex** to utilize rich human hand information efficiently for dexterous manipulation.
- We show the effectiveness of our framework on **12** challenging visual dexterous manipulation tasks, comparing with recent strong foundation models such as VC-1 [17].
- Our study has offered valuable insights into the application of pre-trained models for dexterous manipulation, by exploring the direct application of a 3D human hand pose estimation model, originating from the vision community.

## 2 Related Work

**Visual reinforcement learning for dexterous manipulation.** Recent research has explored the use of deep reinforcement learning (RL) for solving dexterous manipulation tasks [7, 21, 23, 26, 29]. For example, Rajeswaran et al. [23] investigated the use of vector state information as input to the RL algorithm. Despite the success, assuming access to the ground-truth state limits its possibility to be deployed in the real world. RRL [26] finds that ImageNet pre-trained ResNets [10] are surprisingly effective in achieving dexterity with visual observations. Under the umbrella of visual RL, MoDem [7] leverages a learned dynamics model to solve the tasks with good utilization of demonstrations. Furthermore, VRL3 [29] utilizes offline RL to pre-train the visual representations and the policies in an end-to-end manner. In this work, **H-InDex** is designed to focus on visual representations while leaving the policy, training framework, and reward signals unchanged. As a result, **H-InDex** offers an orthogonal and complementary approach to prior efforts in this area.

**Foundation models for visuo-motor control.** Given the diversity of robotic tasks and computational constraints, there is a growing interest in developing a single visual foundation model that can serve as a general feature extractor. Such a model would enable the processing of high-dimensional visual observations into compact vectors, providing a promising approach for efficient and effective control of a wide range of robotic systems [8, 17–19, 26, 30, 32, 33]. Among them, R3M [18] pre-trains a ResNet-50 on Ego4D [6] dataset and evaluates on several robotic manipulation tasks with imitation learning. MVP [30] pre-trains vision transformers [4] with Masked AutoEncoder (MAE) [9] on internet-scale data, achieving strong results on dexterous manipulation tasks. Similarly, a very recent foundation model VC-1 [17] explores the scaling up of MAE for motor control and achieves consistently strong results across a wide range of benchmarks. However, it should be noted that VC-1 and R3M only employ IL to solve dexterous manipulation tasks, making it unclear whether these models are suitable for the setting of reinforcement learning, where agents need to trade off between exploration and exploitation. GNFactor [33] distills the 2D foundation models into 3D space, but their agents are limited to address the gripper-based manipulation problems.

**Learning dexterity from videos.** A growing body of recent research aims to leverage human manipulation videos for improving visuomotor control tasks [17, 20, 22, 27, 28, 30]. A line of works focuses on directly extracting human hand poses from videos and employing RL/IL to train on the retargeted robot joint positions, such as DexMV [22], Robotic telekinesis [28], VideoDex [27], and Imitate Video [20]. In contrast to these approaches, our work explores a representation learning approach for leveraging online human-object interaction videos without explicit pose estimation to improve dexterity in robotic manipulation, sharing a similar motivation as MVP [30] and VC-1 [17].

## 3    Preliminaries

**Formulation.** We model the problem as a Markov Decision Process (MDP) $\mathcal{M} = \langle \mathcal{S}, \mathcal{A}, \mathcal{T}, \mathcal{R}, \gamma \rangle$, where $\mathbf{s} \in \mathcal{S}$ are states, $\mathbf{a} \in \mathcal{A}$ are actions, $\mathcal{T} : \mathcal{S} \times \mathcal{A} \mapsto \mathcal{S}$ is a transition function, $r \in \mathcal{R}$ are rewards, and $\gamma \in [0, 1)$ is a discount factor. The agent's goal is to learn a policy $\pi_\theta$ that maximizes discounted cumulative rewards on $\mathcal{M}$, *i.e.*, $\max_\theta \mathbb{E}_{\pi_\theta} \left[ \sum_{t=0}^\infty \gamma^t r_t \right]$, while using as few interactions with the environment as possible, referred as *sample efficiency*.

In this work, we focus on visual RL for dexterous manipulation tasks, where actions are high-dimensional ($\mathbf{a} \in \mathcal{A}^{30}$) and ground-truth states $s$ are generally unknown, approximated by image observations $\mathbf{o} \in \mathcal{O}$ together with robot proprioceptive sensory information $\mathbf{q} \in \mathcal{Q}$, *i.e.*, $\mathbf{s} = (\mathbf{o}, \mathbf{q})$. To better address the hard exploration problem in high-dimensional control [7, 23, 26, 29], we assume access to a limited number of expert demonstrations $\mathcal{D}_{\text{expert}} = \{D_1, D_2, \cdots, D_N\}$.

**Demo Augmented Policy Gradient** (DAPG) [23] is a model-free policy gradient method that utilizes given demonstrations to augment the policy and trains the policy with natural policy gradient (NPG) [13]. DAPG mainly consists of two stages:

*(1)* Pre-training the policy with behavior cloning, which is to solve the following maximum-likelihood problem:

$$\underset{\theta}{\text{maximize}} \sum_{(s,a) \in \mathcal{D}_{\text{expert}}} \ln \pi_\theta(a \mid s). \tag{1}$$

*(2)* RL finetuning with the demo augmented loss, which is to add an additional term to the gradient,

$$g_{\text{aug}} = \underbrace{\sum_{(s,a) \in \mathcal{D}_\pi} \nabla_\theta \ln \pi_\theta(a \mid s) A^\pi(s,a)}_{\text{original gradient}} + \underbrace{\sum_{(s,a) \in \mathcal{D}_{\text{expert}}} \nabla_\theta \ln \pi_\theta(a \mid s) w(s,a)}_{\text{demo augmented gradient}}, \tag{2}$$

where $A^\pi(s,a)$ is the advantage function, $\mathcal{D}_\pi$ represents the dataset obtained by executing policy $\pi_\theta$ on the MDP, and $w(s,a)$ is a weighting function.

## 4    Method

In this work, our goal is to achieve sample efficient visual reinforcement learning agents in dexterous manipulation tasks by incorporating human hand dexterity into visual representations. To this end, we propose **H**and-**In**formed visual reinforcement learning for **Dex**terous manipulation (**H-InDex**), a simple yet effective learning framework to address the contact-rich dexterous manipulation problems effectively in limited interactions. The overview of our method is provided in Figure 2. H-InDex consists of three stages: *1)* a ***representation pre-training*** stage where we pre-train the visual representations with the 3D human hand pose estimation task, aiming to make visual representations understand human hand dexterity from diverse natural videos; *2)* a ***representation offline adaptation*** stage where we adapt only $0.18\%$ parameters in the pre-trained representation with the self-supervised keypoint objective with in-domain data; *3)* a ***reinforcement learning*** stage where the visual representation is frozen and we utilize the exponential moving average operation to update the mean and variance in BatchNorm of the visual representations.

**Stage 1:** *Representation pre-training.* We start by pre-training visual representations with *monocular 3D human hand pose estimation*, which is a well-established human hand understanding task in the computer vision community with large-scale annotated datasets available. Together with the datasets, there are a plethora of open-sourced models from which we use an off-the-shelf model FrankMocap [24]. FrankMocap is a whole-body pose estimation system with the hand module trained

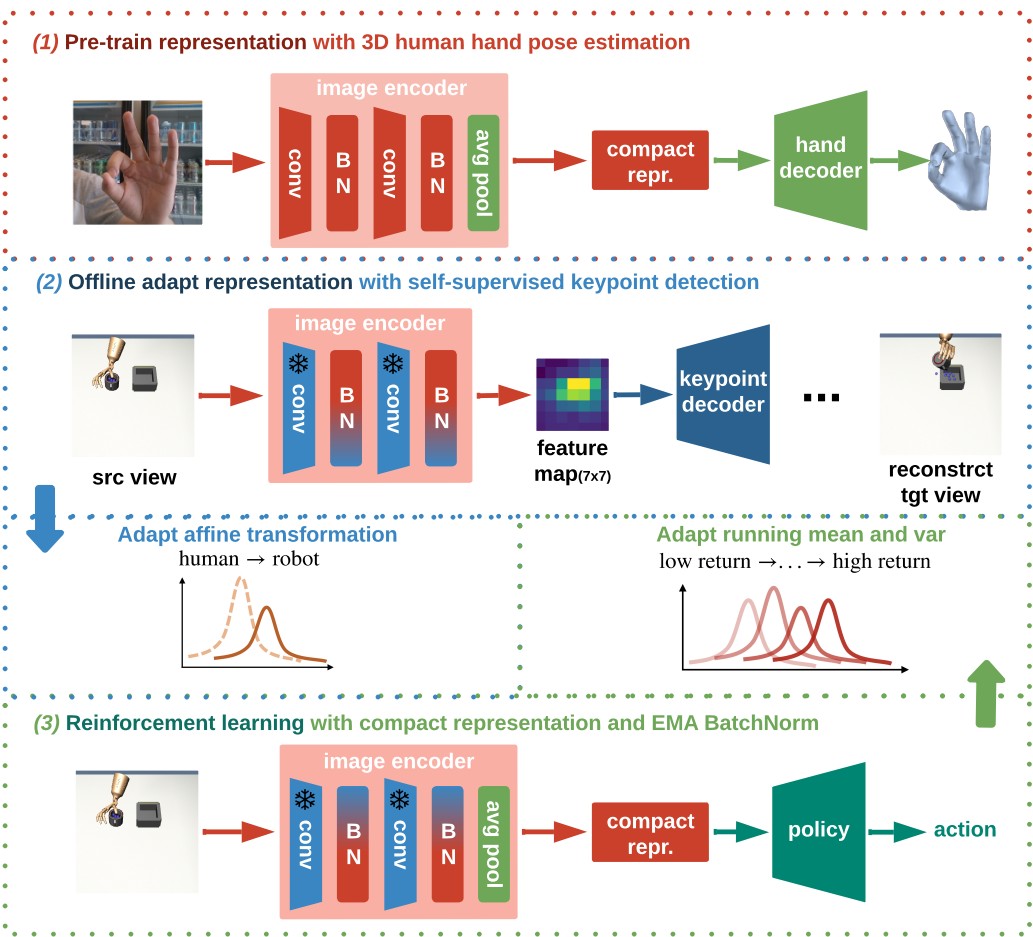

Figure 2: **The overview of H-InDex.** H-InDex consists of three stages: *1)* representation pre-training, *2)* representation offline adaptation, and *3)* reinforcement learning.

on 6 diverse hand datasets, totaling 400k samples. We adopt the ResNet-50 [10] feature encoder in the hand module to extract visual representations.

The use of a pre-trained model from the 3D hand pose estimation task [24] shares the intuition with recent works on foundation models for motor control [16–19, 30]: learning representations from human manipulation videos. However, the use of the pre-trained hand model offers two distinct advantages that are not typically found in other approaches: *i)* the model explicitly predicts the hand-related information from diverse videos, forcing it to learn the interaction and the movement of human hands; *ii)* the model can be borrowed from vision community without any extra cost to re-train a foundation model.

**Stage 2:** *Representation offline adaptation.* In the previous stage, we only pre-train visual representations that are suitable for human-centric images, neglecting the morphology and structure gap between robot hands and human hands. To bridge the gap without losing the information learned in the pre-training stage, we adopt a self-supervised keypoint detection objective [11, 14, 15] to **only** finetune the affine transformations in the BatchNorm layers of the pre-trained model, which occupy only **0.18**% of the entire model parameters. While finetuning only a small portion of parameters, it empirically outperforms both a frozen model and a fully finetuned model. We hypothesize that this is because the BatchNorm finetuning bridges the gap and mitigates catastrophic forgetting caused by finetuning [1].

We now describe the self-supervised keypoint objective. Given a target image $I_t$ and a source image $I_s$ sampled from a video, we aim to reconstruct $I_t$ with the appearance feature of $I_s$ and the keypoint feature of $I_t$. Denote our pre-trained visual representation as $h_\theta$, the keypoint feature extractor as $\mathcal{K}_\psi$, the appearance feature extractor as $\mathcal{F}_\phi$, and the image decoder as $\mathcal{G}_\omega$. First, we extract a semantic

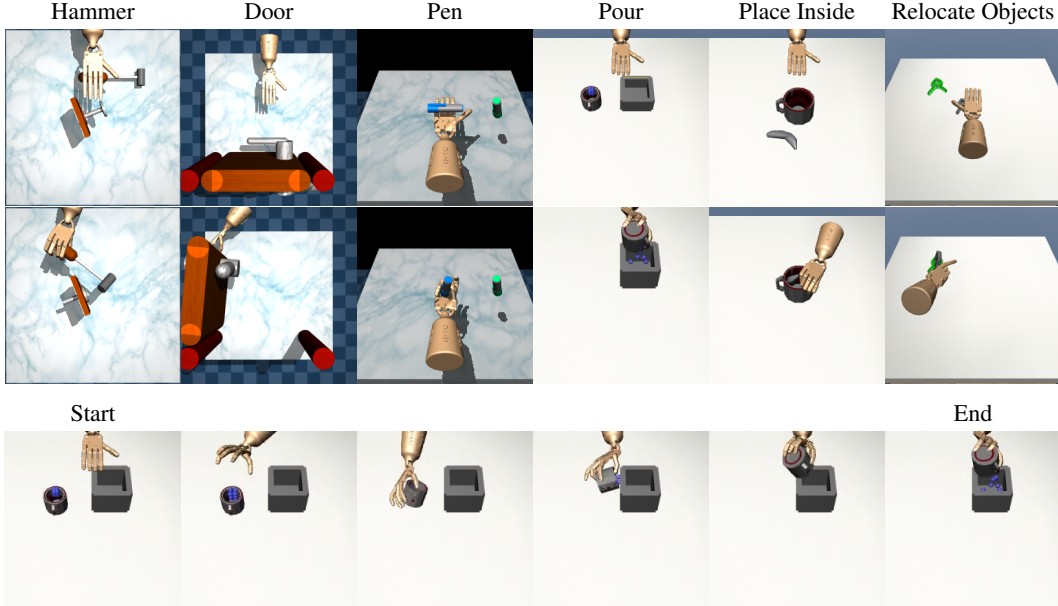

| Hammer | Door | Pen | Pour | Place Inside | Relocate Objects |
|---|---|---|---|---|---|

Start                                                                                                                    End

Figure 3: **Visualization of our six kinds of dexterous manipulation tasks and one sampled trajectory.** We depict both the initial configuration and the goal. Videos of trajectories for all tasks are available on our website **yanjieze.com/H-InDex**.

feature map $h_\theta(I_t)$ from the target image $I_t$ and then get the keypoint feature $\mathcal{K}_\psi(h_\theta(I_t))$. At the same time, we extract the appearance feature $\mathcal{F}_\phi(I_s)$ from the source image $I_s$. We then try to reconstruct the target image $I_t$ by decoding the concatenated keypoint feature and the appearance feature as $I'_t = \mathcal{G}_\omega(\mathcal{K}_\psi(h_\theta(I_t)), \mathcal{F}_\phi(I_s))$. Our final supervision is the perceptual loss [12] $\mathcal{L}_{\text{percep}}$,

$$\mathcal{L}_{\text{keypoint}} = \mathcal{L}_{\text{percep}}(I_t, I'_t) = \|\Lambda(I_t) - \Lambda(\mathcal{G}_\omega(\mathcal{K}_\psi(h_\theta(I_t)), \mathcal{F}_\phi(I_s)))\|_2^2, \tag{3}$$

where $\Lambda$ is the semantic feature prediction function in [12].

**Stage 3:** *Reinforcement learning.* During the reinforcement learning stage, the distribution of observations is continually changing. For example, in the early learning stage, the observations are usually random explorations, while at the end of the learning stage, most of the observations are converged trajectories. Such a property of reinforcement learning requires the internal statistics of neural networks to move slowly towards the current observation distribution. Therefore, we utilize the exponential moving average (EMA) operation to dynamically update the statistics (*i.e.*, the running mean and the running variance) in BatchNorm layers.

Formally, for the input $x$ that has $k$ dimensions, *i.e.*, $x = \{x^{(1)}, \cdots, x^{(k)}\}$, we update the running mean $\mu^{(i)}$ and the running variance $(\sigma^{(i)})^2$ in BatchNorm layers with the following equation,

$$\mu^{(i)} \leftarrow (1 - m) \cdot \mu^{(i)} + m \cdot \mathbb{E}[x^{(i)}], \tag{4}$$

$$(\sigma^{(i)})^2 \leftarrow (1 - m) \cdot (\sigma^{(i)})^2 + m \cdot \text{Var}[x^{(i)}], \tag{5}$$

for $i = 1, \cdots, k$, where $m$ is the momentum. When $m$ is set to 0, our EMA BatchNorm layers revert back to the original layers, ensuring that the modification does not have negative impacts at the very least. Finally, all these three stages collectively contribute to our final method H-InDex. We remain implementation details in Appendix A.

## 5   Experiments

In this work, we delve into the application of visual reinforcement learning to address dexterous manipulation tasks, with a particular emphasis on the visual representation aspect. We evaluate the effectiveness of our proposed framework, H-InDex, across various tasks and elucidate the significance of each component in achieving the final results. Of particular importance is the integration of prior knowledge pertaining to human hand dexterity into our framework.

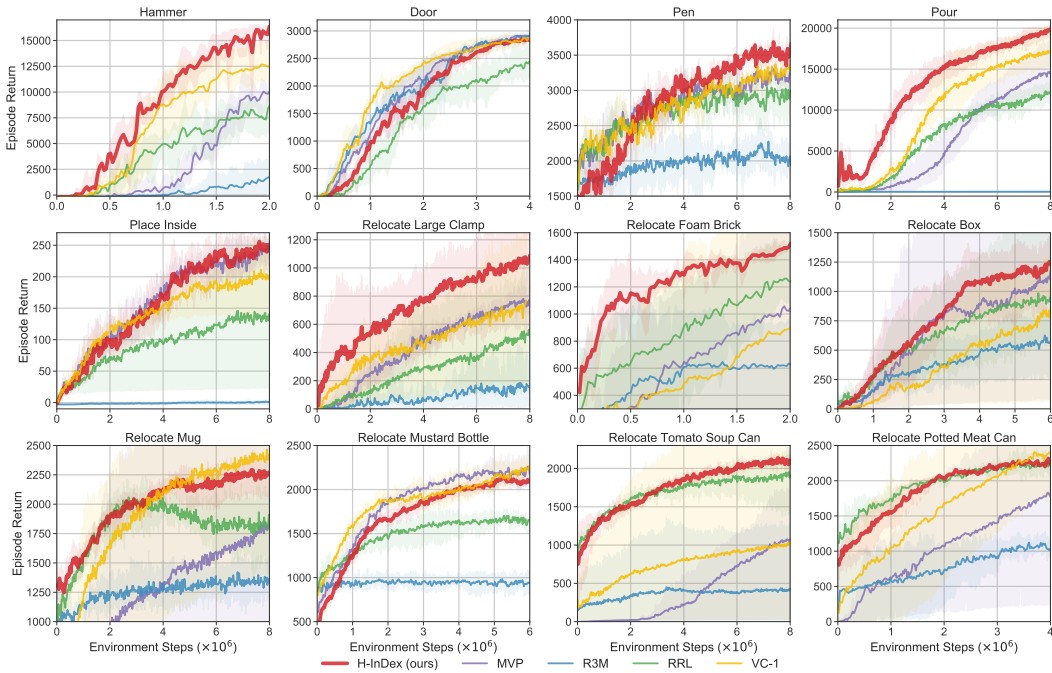

Figure 4: **Episode return for 12 challenging dexterous manipulation tasks.** We compare H-InDex with four strong visual representations for motor control, *i.e.*, VC-1 [17], MVP [30], R3M [18], and RRL [26]. Mean of 3 seeds with seed number $0, 1, 2$. Shaded area indicates 95% confidence intervals.

## 5.1 Experiment Setup

We evaluate H-InDex on **12** challenging visual dexterous manipulation tasks from Adroit [23] and DexMV [22] respectively, including *Hammer*, *Door*, *Pen*, *Pour*, *Place Inside*, and *Relocate YCB Objects* (7 different objects [3]). Visualization of each task is given in Figure 3 and detailed descriptions are provided in Appendix B. This selection of tasks is the most extensive compared to previous works e.g., DAPG (4 tasks) and DexMV (7 tasks), thereby showcasing the robustness and versatility of H-InDex. The tasks were performed with varying numbers of steps based on their level of complexity. We mainly report the cumulative rewards to show the speed of task completion. The dimension of image observations $\mathbf{o} \in \mathcal{O}$ is $3 \times 224 \times 224$ across all methods. We run experiments on an RTX 3090 GPU; each seed takes roughly 12 hours. Due to the limitation on computation resources, we choose to run 3 seeds for each group of experiments and consistently use 3 seeds with seed numbers $0, 1, 2$ to ensure reproducibility. We also observe that H-InDex enjoys a slight variance between seeds, while baselines tend to have a larger variance.

## 5.2 Main Experiments

To demonstrate the effectiveness of H-InDex, we evaluate diverse recent strong visual representations for motor control, including *(i)* VC-1 [17], which trains masked auto-encoders over 5.6M images with over 10,000 GPU-hours and we use the ViT-B [4] model (86M parameters); *(ii)* MVP [30], which also uses masked auto-encoders for pre-training and we use the ViT-S [4] model (22M parameters); *(iii)* R3M [18], which pre-trains a ResNet-50 (22M parameters) with time contrastive learning and video-language alignment *(iv)* RRL [26], which uses the ResNet-50 pre-trained on the ImageNet classification task directly. Due to task differences, we normalize the cumulative rewards based on the highest rewards achieved and present the average scores in Figure 1. We also report the learning curves in Figure 4. We then detail our observations below.

**H-InDex emerges as the dominant representation.** Across 12 tasks, H-InDex outperforms the recent state-of-the-art representation VC-1 by a **16.8%** absolute improvement. Furthermore, H-InDex surpasses RRL, the original state-of-the-art representation in Adroit, by **25.4%**. Analyzing the learning curves, H-InDex demonstrates superior sample efficiency in 10 out of the 12 tasks. In

only two tasks, namely `relocate mug` and `relocate mustard bottle`, VC-1 exhibits a slight advantage over H-InDex.

**ConvNets v.s. ViTs.** Among representations utilizing the ResNet-50 architecture (*i.e.*, H-InDex, R3M, RRL), only H-InDex showcases obvious advantages over ViT-based representations. This suggests that with appropriate domain knowledge, ConvNets can still outperform ViTs. Additionally, we notice that ConvNets and ViTs excel in different tasks. For instance, in `relocate tomato soup can`, VC-1 and MVP achieve returns that are only half of what H-InDex and RRL accomplish. However, in `relocate mug`, VC-1 performs well. These observations highlight the task-dependent nature of the strengths exhibited by ConvNets and ViTs.

### 5.3 The Effectiveness of 3D Human Hand Prior

The significance of transferring the 3D human hand knowledge into dexterous manipulation is non-negligible. To demonstrate the utility of such 3D human hand prior, we compare our vanilla pre-trained representation *i.e.*, the feature extractor from the FrankMocap hand module [24] (denoted as **FrankMocap Hand**) with other 4 representative pre-trained models: *(i)* **FrankMocap Body**, which is the body estimation module from FrankMocap [24], pre-trained with 3D body pose estimation, *(ii)* **AlphaPose** [5], which is a widely-used robust 2D human pose estimation algorithm, *(iii)* **R3M** [18], which is pre-trained with time contrastive learning [25] and language-video alignment on Ego4D [6], and *(iv)* **RRL** [26], which directly uses the ResNet-50 pre-trained on the ImageNet classification task. All the models use a ResNet-50 architecture and do not use any adaptation, ensuring the fairness of our comparison. We also put **H-InDex** as the best results achieved for comparison. Results are shown in Figure 5. We now detail our observations below:

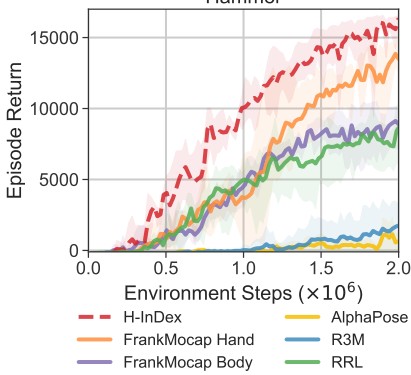

**FrankMocap hand v.s. RRL/R3M.** Our vanilla representation has significantly outperformed both RRL and R3M without the need for any adaptation.

**FrankMocap hand v.s. 3D/2D body-centric representations.** Our vanilla 3D hand representation, FrankMocap Hand, demonstrates superior sample efficiency compared to FrankMocap Body and significantly outperforms AlphaPose. This confirms our hypothesis that the 3D human hand prior is more advantageous than both the 3D and 2D human body prior. It is worth noting that AlphaPose, being a whole-body 2D pose estimation model, is also capable of estimating 2D hand poses. The fact that our 3D hand prior outperforms the 2D hand prior in this context further supports its effectiveness. We hypothesize this is because 2D pose estimation does not require deep spatial reasoning compared to 3D pose estimation.

Figure 5: **Compare vanilla pre-trained representations with H-InDex.**

**H-InDex v.s. FrankMocap hand.** Our vanilla hand representation already surpasses all other pre-trained ConvNets in performance. However, by applying our adaptation technique, we can further enhance the sample efficiency of the hand representation, underscoring the significance of adapting the model with in-domain data in a proper way.

### 5.4 Ablations

To validate the rationale behind the design choices of H-InDex, we performed a comprehensive set of ablation experiments.

**Effects of each stage.** Figure 6a provides insights into the contributions of each stage towards the overall efficiency of H-InDex. We refer to RRL as **w/o Stage 1,2,3**. Significantly, Stage 1 exhibits the most notable enhancement, underscoring the efficacy of human dexterity. Moreover, Stage 2 and Stage 3 also contribute appreciable advancements. In addition, the value of momentum ($m$) in Stage 3 has a significant influence, as illustrated in Figure 6b. To determine the optimal value, we performed a grid search over $m \in \{0, 0.1, 0.01, 0.001\}$. This analysis highlights the importance of selecting an appropriate momentum value for achieving optimal performance in Stage 3 of H-InDex.

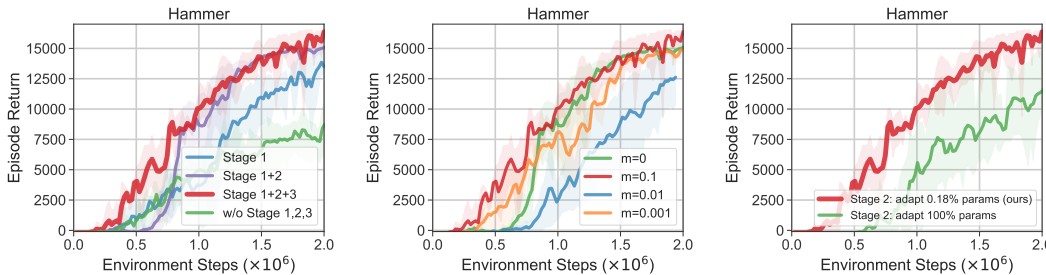

(a) Ablation on the effectiveness of our three stages.

(b) Ablation on momentum $m \in \{0, 0.1, 0.01, 0.001\}$.

(c) Compare our minimal adaptation in Stage 2 and the full adaptation.

Figure 6: **Ablation experiments.** We ablate each component of H-InDex and show that each individual part effectively combines to contribute to the overall effectiveness of H-InDex.

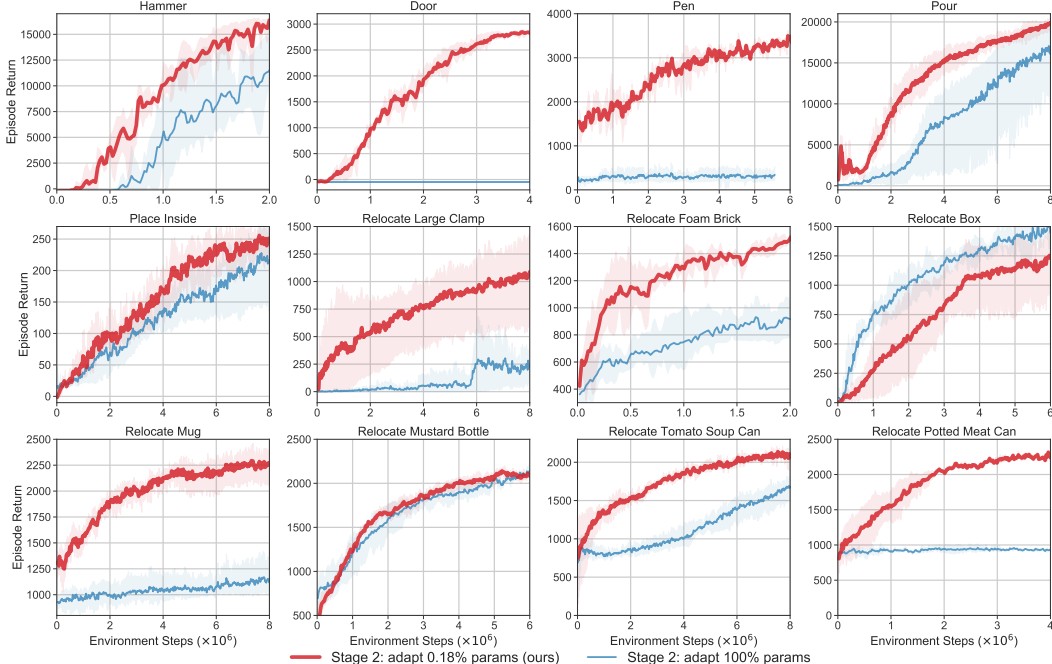

Figure 7: **Ablation on Stage 2 (adapting $0.18\%$ parameters or adapting $100\%$ parameters).** We observe that simply finetuning the entire visual representation would lead to sub-optimal results. Instead, H-InDex only adapts the parameters in BatchNorm layers and effectively solves all the tasks.

Figure 9 provides more ablation results on Stage 3, further supporting the necessity of updating the BatchNorm layers during the training of RL agents.

**Adapting $100\%$ parameters v.s. adapting $0.18\%$ parameters in Stage 2.** In Stage 2 of H-InDex, we intentionally chose to adapt only $0.18\%$ of the parameters (the affine transformations in BatchNorm layers) in the pre-trained representation. This decision was made to address a specific concern, as depicted in Figure 6c and Figure 7. By altering only the setting for Stage 2 while keeping all other factors constant, we observed that across all tasks, adapting all parameters is not more advantageous. This phenomenon may be attributed to the fact that freezing and finetuning partial parameters help mitigate catastrophic forgetting, which is often caused by full finetuning [1]. In Figure 8, we also show the necessity of our Stage 2. We could conclude from results that correctly finetuning the visual representation is one key to the stable convergence, and not correctly finetuning the model, such as finetuning all the parameters, could be even worse than the frozen model.

**Robust visual generalization.** One concern of our hand representation is its generalization ability, compared to the vision model pre-trained on large-scale datasets, such as VC-1 [17]. Therefore, we change the background of the training scene to various novel backgrounds, as shown in Figure 12 (see Appendix C) and evaluate VC-1 and H-InDex on the task `relocate potted meat can`. The

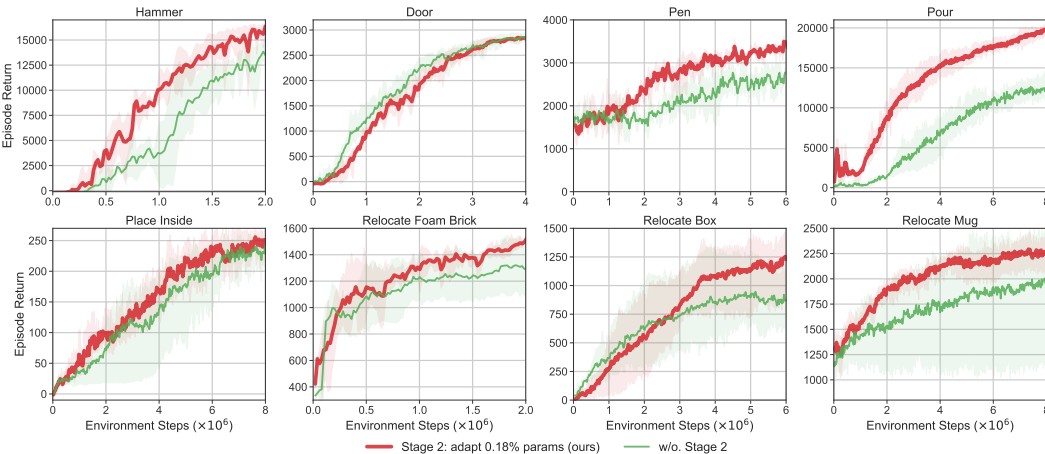

Figure 8: **Ablation on Stage 2 (with or without adaptation).** We also conduct more experiments to show the necessity of Stage 2. We could observe a consistent improvement across these tasks by applying Stage 2, which only adapts $0.18\%$ parameters of the visual representation.

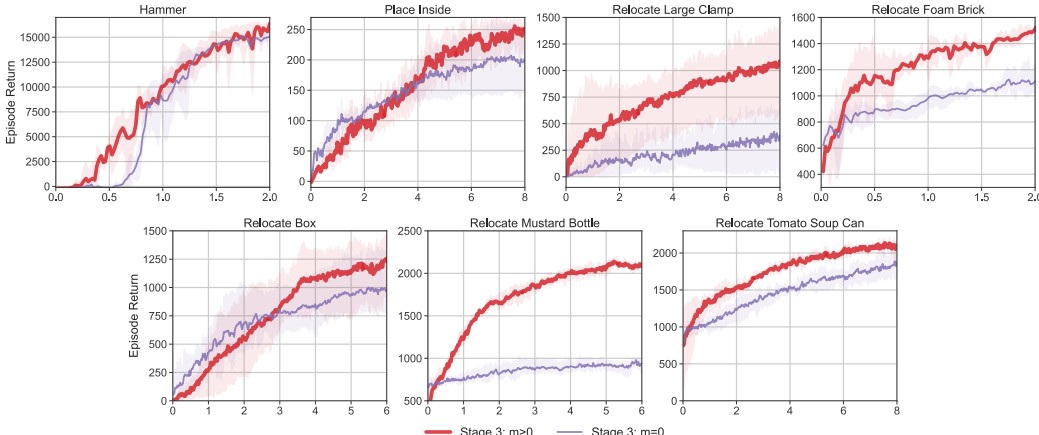

Figure 9: **Ablation on Stage 3 (momentum $m > 0$ or $m = 0$).** We observe that our Stage 3 contributes greatly to some specific tasks, such as `relocate mustard bottle`, and for some tasks like `hammer`, tuning this parameter only results in a slightly faster convergence. For tasks not shown here, we all use $m = 0$, since H-InDex with Stage 1 and Stage 2 has been strong enough.

results given in Table 3 (see Appendix C) show that H-InDex could handle the changed background better than VC-1. We also see the consistent performance drop across all scenes, emphasizing the importance of visual generalization.

**Visualization of self-supervised keypoint detection in Stage 2.** In Stage 2, a self-supervised keypoint detection objective is employed to fine-tune a minimal percentage $(0.18\%)$ of parameters in the pre-trained model. The visualization results, as shown in Figure 10, demonstrate the successful detection of keypoints. This observation highlights the pre-trained model's capability to effectively allocate attention to the hand and objects depicted in the images, even with the adaptation of only a small subset of parameters.

**Visualization of affine transformations adaptation in Stage 2.** The significance of Stage 2 in H-InDex is evident from Figure 6a. To gain deeper insights into this phenomenon, we visualize the distribution of the adapted parameters, specifically the affine transformations in BatchNorm layers. We accomplish this by fitting a Gaussian distribution. Figure 11 presents the visualization results, highlighting an interesting trend. For the shallow layers, the distributions of the adapted models closely resemble those of the pre-trained models. However, as we move deeper into the layers, noticeable differences emerge. We attribute this disparity to the fact that dissimilarities between human hands and robot hands extend beyond low-level features like color and texture. Instead,

| Hammer | Pen | Pour | Relocate Objects |

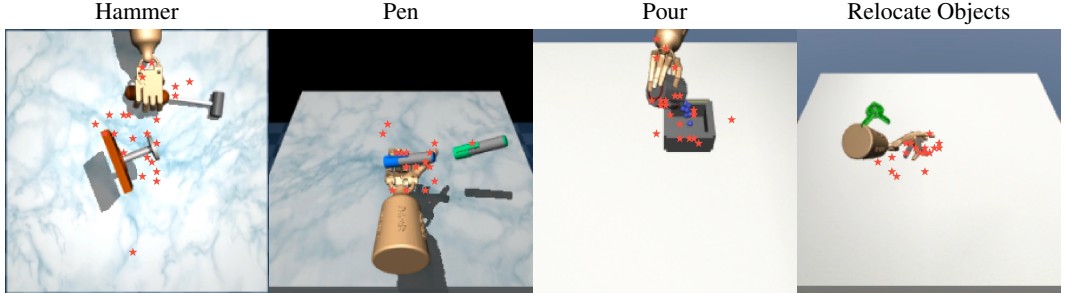

Figure 10: **Visualization of our self-supervised keypoint detection.** We select four tasks here and mark the detected keypoints on images with ⋆red stars. We observe that these keypoints consistently mark the dynamic regions of images. Full videos are available on **yanjieze.com/H-InDex**.

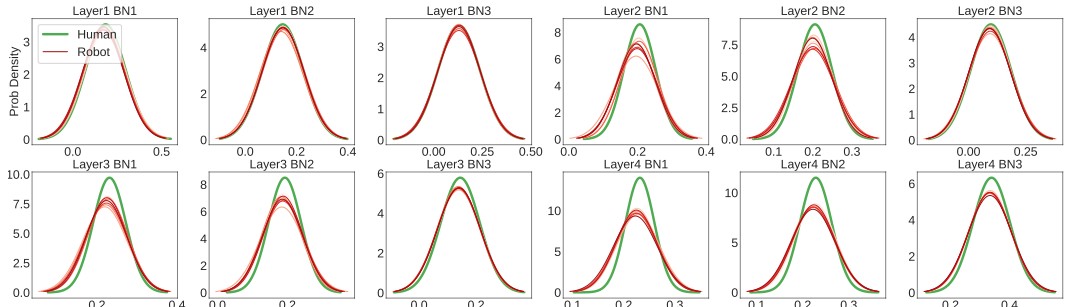

Figure 11: **Visualization of affine transformation adaptation in Stage 2.** We fit a Gaussian distribution for parameters of affine transformations. We omit the X-axis and Y-axis here for simplicity, Human represents the original pre-trained model and robot represents the adapted representation in different tasks. It is observed that deep layers are subjected to large distribution shifts.

they encompass higher-level features such as dynamics and structure [2, 31, 34]. This observation underscores the importance of our adaptation approach, as it effectively addresses the variations in both low-level and high-level features, facilitating the success of H-InDex.

## 6 Conclusion

In this study, we introduce H-InDex, a visual reinforcement learning framework that leverages hand-informed visual representations to tackle complex dexterous manipulation tasks effectively. H-InDex outperforms other recent state-of-the-art representations in a range of 12 tasks, including six kinds of manipulation skills. The effectiveness of H-InDex can be attributed to its three-stage approach, wherein Stage 1 incorporates a pre-trained 3D human hand representation and Stage 2 and Stage 3 focus on careful in-domain adaptation with only $0.36\%$ parameters updated. These stages collectively contribute to the successful preservation and utilization of the human hand prior knowledge.

It is also important to acknowledge some limitations of our work. We did not investigate the generalization capabilities of H-InDex, particularly in scenarios involving the grasping of novel objects. Our future work aims to address this limitation and enhance H-InDex's generalization capabilities for real-world applications. In addition, we find that our Stage 3 is surprisingly effective in some specific tasks like `relocate mustard bottle`, while we have not given a theoretical understanding of such phenomena. We consider this problem as a possible future direction.

## Acknowledgment

This work is supported by National Key R&D Program of China (2022ZD0161700).

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

# Appendix

## A Implementation Details

**Codebase.** Our major codebase is built upon the official implementation of RRL [26], which is publicly available on https://github.com/facebookresearch/RRL and includes the Adroit manipulation tasks [23]. The DexMV [22] tasks are from the official code https://github.com/yzqin/dexmv-sim. All the visual representations in our work are also available online, including RRL (pre-trained ResNet-50, provided in PyTorch officially), R3M (https://github.com/facebookresearch/r3m), MVP (https://github.com/ir413/mvp), VC-1 (https://github.com/facebookresearch/eai-vc), and FrankMocap (https://github.com/facebookresearch/frankmocap). This ensures the good reproducibility of our work. *We are also committed to releasing the code.*

**Network architecture for H-InDex.** The architecture employed by H-InDex is based on ResNet-50 [10], referred to as $h_\theta$. In the initial stage (Stage 1), $h_\theta$ takes as input a $224 \times 224$ RGB image and processes it to generate a compact vector of size $2048$. In Stage 2, we modify $h_\theta$ by removing the average pooling layer in the final layer, resulting in the image being decoded into a feature map with dimensions $7 \times 7 \times 2048$. Moving on to Stage 3, $h_\theta$ once again produces a compact vector of size $2048$, while simultaneously updating the statistics within the BatchNorm layers using the exponential moving average operation.

**Implementation details for Stage 2.** Our implementation strictly follows the previous work that also uses the self-supervised keypoint detection as objective [11,14,15]. We give a PyTorch-style overview of the learning pipeline below and refer to [11] for more implementation details. Notably, the visual representation $h_\theta$ (24M) contains the majority of parameters, while all other modules in the pipeline maintain a parameter count ranging from 1M to 3M. We use 50 demonstration videos as training data for each task and train 100k iterations to ensure convergence with learning rate $1 \times 10^{-4}$. One of our core contributions is to only adapt the parameters in BatchNorm layers in $h_\theta$, and we emphasize that the learning objective is not our contribution, as it has been well explored in [11,14,15].

```
for _ in range(num_iters):
    # sample data
    source_view, target_view = next(data_iter) # 3x224x224

    # self-supervised keypoint-based reconstruction
    # h_theta is our visual representation
    feature_map = h_theta(target_view) # -> 7x7x2048
    keypoint_feat = keypoint_encoder(feature_map) # -> 30x56x56
    keypoint_feat = up_sampler(keypoint_feature) # -> 256x28x28
    apperance_feat = apperance_encoder(source_view) # -> 256x28x28
    target_view_recon = image_decoder([keypoint_feat,apperance_feat]) # -> 3x224x224

    # compute loss
    loss = perceptual_loss(target_view, target_view_recon)

    # compute gradient and update model
    optimizer.zero_grad()
    loss.backward()
    optimizer.step()
```

## B Task Descriptions

In this section, we briefly introduce our tasks. We use an Adroit dexterous hand for manipulation tasks. The task design follows Adroit [23] and DexMV [22]. Visualizations of task trajectories are available at h-index-rl.github.io.

**Hammer (Adroit).** It requires the robot hand to pick up the hammer on the table and use the hammer to hit the nail.

**Door (Adroit).** It requires the robot hand to open the door on the table.

**Pen (Adroit).** It requires the robot hand to orient the pen to the target orientation.

**Pour (DexMV).** It requires the robot hand to reach the mug and pour the particles inside into a container.

**Place inside (DexMV).** It requires the robot hand to place the object on the table into the mug.

**Relocate YCB objects [3] (DexMV).** It requires the robot hand to pick up the object on the table to the target location. The objects in our tasks include *foam brick*, *box*, *mug*, *mustard bottle*, *tomato soup can*, and *potted meat can*.

## C  Visual Generalization

One concern of our hand representation is its generalization ability, compared to the vision model pre-trained on large-scale datasets, such as VC-1 [17]. Therefore, we change the background of the training scene to various novel backgrounds, as shown in Figure 12 and evaluate VC-1 and H-InDex on the task `relocate potted meat can`. The results given in Table 3 show that H-InDex could handle the changed background better than VC-1. We also see the consistent performance drop across all scenes, emphasizing the importance of visual generalization.

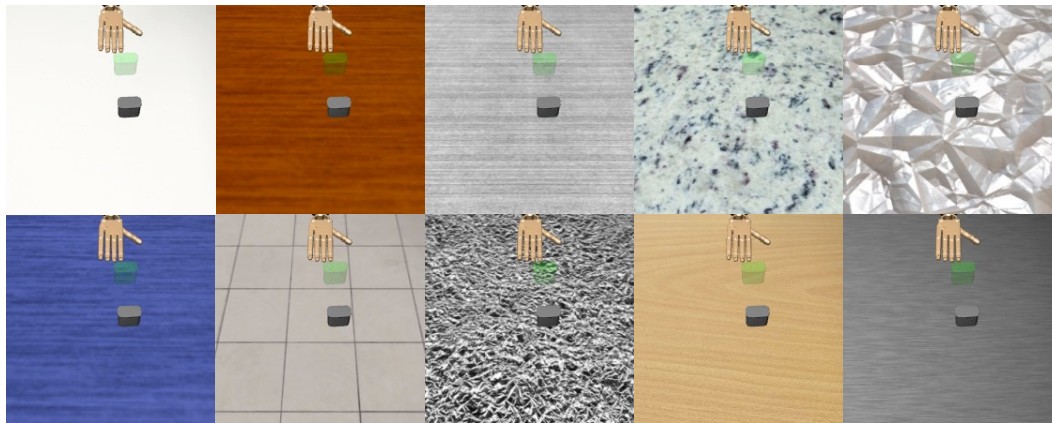

Figure 12: **Various backgrounds for visual generalization.** The first image shows the training scene and the rest 9 images show the novel scene.

Table 1: **Scores for generalization to unseen backgrounds** on `relocate potted meat can` task. We evaluate VC-1 and H-InDex with 20 episodes for each seed.

| Scene ID / Method | VC-1 [17] | H-InDex |
|:---:|:---:|:---:|
| Origin | $\mathbf{2391.74_{\pm 602.83}}$ | $2240.37_{\pm 85.45}$ |
| 1 | $896.28_{\pm 1006.55}$ | $\mathbf{915.95_{\pm 922.65}}$ |
| 2 | $603.26_{\pm 920.48}$ | $\mathbf{771.28_{\pm 793.51}}$ |
| 3 | $451.36_{\pm 839.45}$ | $\mathbf{578.42_{\pm 764.09}}$ |
| 4 | $360.21_{\pm 772.64}$ | $\mathbf{472.66_{\pm 715.03}}$ |
| 5 | $300.02_{\pm 718.05}$ | $\mathbf{393.32_{\pm 676.41}}$ |
| 6 | $256.80_{\pm 673.16}$ | $\mathbf{340.07_{\pm 639.68}}$ |
| 7 | $224.21_{\pm 635.56}$ | $\mathbf{298.20_{\pm 608.54}}$ |
| 8 | $226.59_{\pm 610.58}$ | $\mathbf{265.82_{\pm 581.03}}$ |
| 9 | $214.30_{\pm 581.76}$ | $\mathbf{239.60_{\pm 556.80}}$ |
| Average | 392.56 | 475.04 |

## D  Main Experiments (ConvNets Only)

In our primary experimental analysis, we conduct a comprehensive comparison of five visual representations, with three of them being ConvNets, including our method. Figure 13 presents an isolated

demonstration of the comparison among the ConvNets. Notably, our method H-InDex exhibits superior performance in comparison to the other ConvNets.

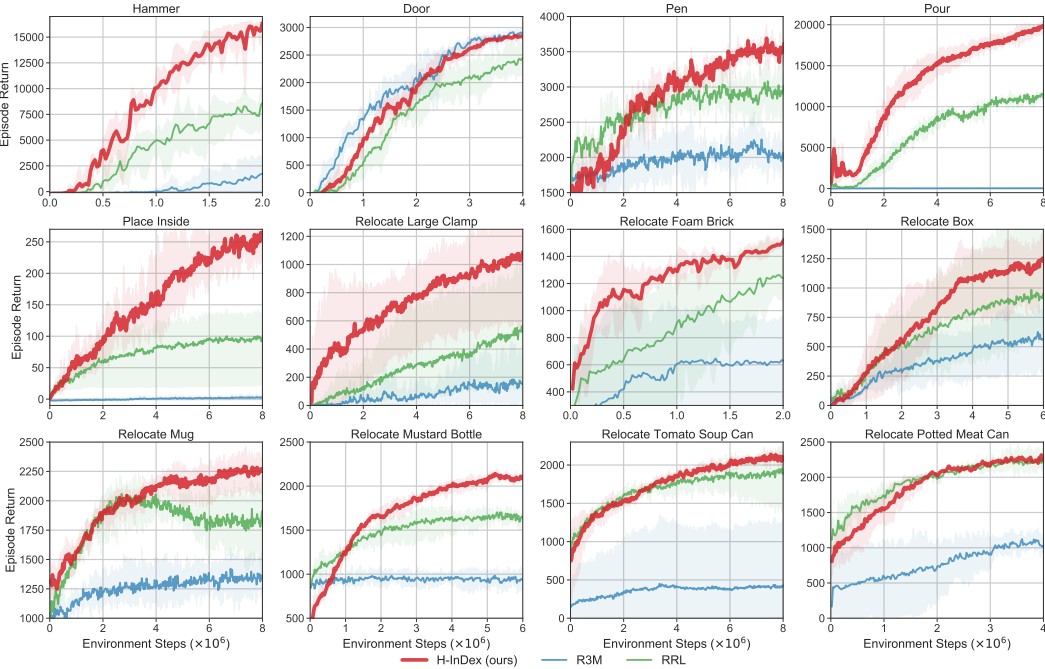

Figure 13: Episode return for **12** challenging dexterous manipulation tasks. Mean of 3 seeds with seed number $0, 1, 2$. Shaded area indicates 95% CIs.

# E   Success Rates in Main Experiments

We present the success rates of our six task categories as in Table 3. Regarding the hammer task, it is evident that both H-InDex and VC-1 exhibit success rates near $100\%$. However, a notable disparity arises when considering episode returns, indicating the varying degrees of task execution proficiency even among successful agents.

Table 2: **Success rates for main experiments.** Highest success rates for each task are marked with **bold** fonts. The success rates only reflect whether the task is finished but do not reflect how fast the task is finished. H-InDex still dominates other methods.

| Task name / Method | RRL [26] | R3M [18] | MVP [30] | VC-1 [17] | H-InDex |
|---|---|---|---|---|---|
| Hammer (2M) | $89_{\pm15}$ | $24_{\pm21}$ | $83_{\pm11}$ | $97_{\pm3}$ | $\mathbf{100_{\pm0}}$ |
| Door (4M) | $92_{\pm1}$ | $99_{\pm2}$ | $\mathbf{100_{\pm0}}$ | $99_{\pm2}$ | $96_{\pm5}$ |
| Pen (8M) | $78_{\pm4}$ | $58_{\pm6}$ | $80_{\pm4}$ | $81_{\pm2}$ | $\mathbf{90_{\pm2}}$ |
| Pour (8M) | $38_{\pm33}$ | $0_{\pm0}$ | $23_{\pm38}$ | $67_{\pm29}$ | $\mathbf{99_{\pm2}}$ |
| Place inside (6M) | $70_{\pm50}$ | $1_{\pm2}$ | $\mathbf{99_{\pm2}}$ | $98_{\pm4}$ | $97_{\pm6}$ |
| Relocate large clamp (8M) | $33_{\pm31}$ | $0_{\pm0}$ | $47_{\pm27}$ | $23_{\pm21}$ | $\mathbf{50_{\pm45}}$ |
| Relocate foam brick (2M) | $\mathbf{87_{\pm11}}$ | $42_{\pm37}$ | $48_{\pm46}$ | $44_{\pm49}$ | $86_{\pm10}$ |
| Relocate box (6M) | $94_{\pm5}$ | $45_{\pm24}$ | $48_{\pm50}$ | $49_{\pm50}$ | $85_{\pm14}$ |
| Relocate mug (2M) | $\mathbf{100_{\pm0}}$ | $82_{\pm2}$ | $54_{\pm51}$ | $74_{\pm44}$ | $\mathbf{100_{\pm0}}$ |
| Relocate mustard bottle (2M) | $\mathbf{100_{\pm0}}$ | $82_{\pm8}$ | $\mathbf{100_{\pm0}}$ | $99_{\pm2}$ | $99_{\pm2}$ |
| Relocate tomato soup can (2M) | $97_{\pm3}$ | $18_{\pm31}$ | $6_{\pm10}$ | $30_{\pm52}$ | $\mathbf{99_{\pm2}}$ |
| Relocate potted meat can (2M) | $\mathbf{97_{\pm6}}$ | $56_{\pm15}$ | $69_{\pm48}$ | $88_{\pm21}$ | $94_{\pm5}$ |
| Average | $81.3$ | $42.3$ | $63.1$ | $70.8$ | $91.3$ |

Table 3: **Success rates and scores for generalization to unseen backgrounds** on `relocate potted meat can` task. We evaluate VC-1 and H-InDex with 20 episodes for each seed.

| Scene ID / Method | VC-1 [17] (success rate) | VC-1 [17] (score) | H-InDex (success rate) | H-InDex (score) |
|---|---|---|---|---|
| 1 | $38_{\pm43}$ | $896.28_{\pm1006.55}$ | $48_{\pm48}$ | $915.95_{\pm922.65}$ |
| 2 | $26_{\pm39}$ | $603.26_{\pm920.48}$ | $32_{\pm46}$ | $771.28_{\pm793.51}$ |
| 3 | $19_{\pm36}$ | $451.36_{\pm839.45}$ | $24_{\pm42}$ | $578.42_{\pm764.09}$ |
| 4 | $15_{\pm33}$ | $360.21_{\pm772.64}$ | $19_{\pm39}$ | $472.66_{\pm715.03}$ |
| 5 | $13_{\pm31}$ | $300.02_{\pm718.05}$ | $16_{\pm36}$ | $393.32_{\pm676.41}$ |
| 6 | $11_{\pm29}$ | $256.80_{\pm673.16}$ | $14_{\pm34}$ | $340.07_{\pm639.68}$ |
| 7 | $10_{\pm27}$ | $224.21_{\pm635.56}$ | $12_{\pm32}$ | $298.20_{\pm608.54}$ |
| 8 | $10_{\pm26}$ | $226.59_{\pm610.58}$ | $11_{\pm30}$ | $265.82_{\pm581.03}$ |
| 9 | $9_{\pm25}$ | $214.30_{\pm581.76}$ | $10_{\pm29}$ | $239.60_{\pm556.80}$ |
| Average | 16.8 | 392.56 | 20.7 | 475.04 |

# F  Hyperparameters

We categorize hyperparameters into task-specific ones (Table 4) and task-agnostic ones (Table 5), Across all baselines, all the hyperparameters are shared except the momentum $m$, which is only used in our algorithm. All the hyperparameters for policy learning are the same as RRL [26]. This ensures the comparison between different representations is fair.

Our exploration of the momentum $m$ in Table 4 has been limited to a specific set of values, namely $\{0, 0.1, 0.01, 0.001\}$, through the use of a grid search technique, due to the limitation on computation resources. It is observed that carefully tuning $m$ could take more benefits.

Table 4: **Task-specific hyperparameters.**

| Task name / Variable | Momentum $m$ | Demonstrations | Training steps (M) | Episode length |
|---|---|---|---|---|
| Hammer | 0.1 | 25 | 2 | 200 |
| Door | 0.0 | 25 | 4 | 200 |
| Pen | 0.0 | 25 | 6 | 100 |
| Pour | 0.0 | 50 | 8 | 200 |
| Place inside | 0.001 | 50 | 8 | 200 |
| Relocate large clamp | 0.01 | 50 | 8 | 100 |
| Relocate foam brick | 0.01 | 25 | 2 | 100 |
| Relocate box | 0.001 | 25 | 6 | 100 |
| Relocate mug | 0.0 | 25 | 8 | 100 |
| Relocate mustard bottle | 0.001 | 25 | 6 | 100 |
| Relocate tomato soup can | 0.01 | 25 | 8 | 100 |
| Relocate potted meat can | 0.0 | 25 | 4 | 100 |

Table 5: **Task-agnostic hyperparameters.**

| Variable | Value |
|---|---|
| Dimension of image observations | $224 \times 224 \times 3$ |
| Dimension of robot states | 30 |
| Dimension of actions | 30 |
| Hidden dimensions of policy $\pi$ | $256, 256$ |
| BC learning rate | 0.001 |
| BC epochs | 5 |
| BC batch size | 32 |
| RL learning rate | 0.001 |
| Number of trajectories for one step | 100 |
| VF batch size | 64 |
| VF epochs | 2 |
| RL step size | 0.05 |
| RL gamma | 0.995 |
| RL gae | 0.97 |

