# OpenReview forum: "H-InDex: Visual Reinforcement Learning with Hand-Informed Representations for Dexterous Manipulation"
_NeurIPS.cc/2023/Conference — NeurIPS 2023 poster_

### Official Review · Reviewer_di2q · 2023-06-26

**Soundness:** 2 fair
**Presentation:** 3 good
**Contribution:** 3 good
**Rating:** 5
**Confidence:** 3

**Summary:**

The paper works on the problem of visual RL dexterous manipulation with pre-trained representations. Specifically, the authors propose to fix the pre-trained Convolution layers and adapt only the BatchNorm layers on a small amount of in-domain data, and then learn the policy based on the pre-trained feature with EMA updates. Experiments are performed on 12 simulation tasks.

**Strengths:**

The idea of leveraging large-scale pretraining to boost the performance of visual RL is interesting and well-motivated;

The proposed method for adapting pre-trained feature for dexterous manipulation is simple and straightforward;

The paper is well-written and easily read.

**Weaknesses:**

For some tasks, the proposed method seems on par or slightly worse than baselines (e.g., Door, Relocate Mug, Relocate Mustard Bottle);

The necessity of stage 3 in the pipeline can be called into question since the performance of the pipeline without stage 3 appears to be comparable to the performance of the full pipeline in the Hammer task.

No limitations or failure modes are presented in the paper.

**Questions:**

In Fig. 6 (b), why would larger or smaller momentums lead to better performance than an intermediate one? Could authors give more theoretical analysis on this?

It seems like the standard BatchNorm without EMA already achieves comparable performance. It would be great if more theoretical guidance on choosing the momentum value is provided instead of using empirical study;

Further clarification or analysis from the authors regarding the significance and contribution of stage 3 would be valuable to better understand its role in achieving improved results;

In Fig. 8, it would be better to also present the scales for X-axis and Y-axis.

**Limitations:**

No limitations or failure modes are presented in the paper.

---

> ### Author Rebuttal · Authors · 2023-08-09
>
> We thank the reviewer for the constructive comments and suggestions. We address each of your comments in the following.
>
> **Q1:** For some tasks, the proposed method seems on par or slightly worse than baselines (e.g., Door, Relocate Mug, Relocate Mustard Bottle);
>
> **A1:** In the field of deep reinforcement learning, it is inherently challenging for a single approach to excel across all tasks. In our experiments, although VC-1 slightly outperforms our method in tasks like Relocate Mug and Relocate Mustard Bottle, it significantly lags behind in others, including Relocate Tomato Soup Can and Relocate Foam Brick. We agree with the reviewer that the pursuit of a universally optimal representation is critical. Nevertheless, our method delivers robust performance across a spectrum of tasks, only equating to baseline levels in a few tasks. We believe the results demonstrate the advantage on both task quantities and average performance.
>
> **Q2:** The necessity of stage 3 in the pipeline can be called into question since the performance of the pipeline without stage 3 appears to be comparable to the performance of the full pipeline in the Hammer task.
>
> **A2:** To further show the functionality of our Stage 3, we conduct a more comprehensive evaluation during the rebuttal phase. Our experiments are conducted across all 7 tasks where the momentum m > 0. The results are given in Figure 3 of the rebuttal PDF file. It is observed that our simple adaptation in Stage 3 is helpful for all these tasks, especially for Relocate Mustard Bottle and Relocate Large Clamp. In our initial submission, due to the computation limit, we do not ablate on all the tasks and only put results on hammer.
>
> **Q3:** No limitations or failure modes are presented in the paper.
>
> **A3:** We have described the limitations in the conclusion section. Our framework also has failure modes. For instance, in Figure 2 of the rebuttal file, on Relocate Box task, removing Stage 2 proves to be more effective than incorporating it. This suggests that merely combining Stage 1 with Stage 3 might be a sufficiently effective approach.
>
> **Q4:** In Fig. 6 (b), why would larger or smaller momentums lead to better performance than an intermediate one? Could authors give more theoretical analysis on this?
>
> **A4:** Intuitively, the larger momentum means that the model values the current and recent data points more and assigns them a larger weight, and this could be suitable to the case where the policy learns the tasks very quickly and thus the incoming data is changing quickly. In our experiments, the methods converge fast on *hammer* compared to other tasks, and thus the momentum might be large (m=0.1). Following these intuitions, we set the momentum by additional considering empirical performance. It could be good if the rigorous theoretical analysis is built and the momentum could be selected based on the theory, which is not the focus of  work but could be an interesting problem for future research.
>
> **Q5:** It seems like the standard BatchNorm without EMA already achieves comparable performance. It would be great if more theoretical guidance on choosing the momentum value is provided instead of using empirical study;
>
> **A5:** See A2 and A4.
>
> **Q6:** Further clarification or analysis from the authors regarding the significance and contribution of stage 3 would be valuable to better understand its role in achieving improved results;
>
> **A6:** See A2.
>
> **Q7:** In Fig. 8, it would be better to also present the scales for X-axis and Y-axis.
>
> **A7:** We update Figure 8 with X and Y axis. The new figure is also given as Figure 5 of the rebuttal file. Thank the reviewer for this suggestion.
>
> **Please do not hesitate to let us know if you have any additional comments.**

---

> > ### Comment · Reviewer_di2q · 2023-08-18
> > **POST-REBUTTAL**
> >
> > I extend my gratitude to the authors for sharing their feedback. The additional results and analysis effectively alleviate my prior concerns regarding the technical details of the methodology.

---

### Official Review · Reviewer_rofX · 2023-07-03

**Soundness:** 3 good
**Presentation:** 3 good
**Contribution:** 4 excellent
**Rating:** 6
**Confidence:** 4

**Summary:**

The paper proposes H-InDex for dexterous manipulation tasks with a 30 DoF robotic hand in simulation. The method consists of three stages: the first stage adopts a pre-trained network that is trained with a 3D human hand pose estimation dataset; the second stage adapts the representation with self-supervised keypoint detection, which updates only 0.18\% parameters of the entire model; and the final stage learns an RL policy on top of the adapted module while also adapting representation by using the exponential moving average for changing distribution based on training state.

The method is extensively evaluated on 12 different visual dexterous manipulation tasks and is demonstrated it outperforms the current SoTA baselines by a 16.8\% absolute improvement and also achieves superior sample efficiency in 10 out of the 12 tasks. The authors also tested the effectiveness of the proposed human hand prior by comparing the method with several different priors and showed the hand prior from human movies does improve the performance. Finally, ablations are conducted to show the effects of each state, how the adaptation works, qualitative results on keypoint detection, and visualization of the adaptation in BatchNorm layers.

Overall, the paper is very well written and organized, and the reviewer enjoyed reading this work. However, there are some unclear descriptions and conclusions that cannot be drawn from the experiments. The reviewer would be happy to raise the score if the authors could answer the questions and/or clarify the reviewer's misunderstanding.

**Strengths:**

- The idea of using a prior from human hand pose estimation is interesting, and the experimental results support the soundness of the method.
- Only adapting the limited parameters by changing the statistics of the BatchNorm layers is also interesting and could be useful for other tasks that involve domain transfer, such as sim2real.
- The experiments are extensively conducted and they support the idea of the proposed method. However, there are some unclear descriptions and/or conclusions that are summarized in the Weaknesses section.

**Weaknesses:**

- The authors draw some conclusions that are not empirically supported or require more experiments. The reviewer thinks the authors need to tone down their claims and/or cite adequate prior works to justify the claims. For example,
  - L.145-146: 'While finetuning only a small portion of parameters, it empirically outperforms both a frozen model and a fully finetuned model.' Please show the numbers probably in the appendix.
  - L.209,212: 'with appropriate domain knowledge, ConvNets can still outperform ViT', 'these observations highlight the task-dependent nature of the strengths exhibited by ConvNets and ViTs.' The reviewer thinks these are rather the nature of algorithms, not feature extractors (which is also just a hypothesis, without any evidence). To draw this conclusion, the authors need to do experiments with the same backbone networks, the same algorithms, and large enough amounts of random seeds. Without those, the reviewer does not think these claims are valid.
  - L.273-276: 'For the shallow layers, the distributions of the adapted models closely resemble those of the pre-trained models. However, as we move deeper into the layers, noticeable differences emerge. We attribute this disparity to the fact that dissimilarities between human hands and robot hands extend beyond low-level features like color and texture. Instead, they encompass higher-level features such as dynamics and structure.' At the very least, please cite relevant papers that analyze the adaptation between two domains. Especially, it would be very difficult to claim that 'they encompass higher-level features such as dynamics and structure.'. For what reasons did the authors draw this conclusion?
- Fig.7: It is very hard to understand whether the keypoints are reasonable or not. For example, the bottom red star of the leftmost figure does not look meaningful.
- Fig.6: Why do the authors specifically choose the three environments, specifically why only (c) is conducted in the 'Place Inside'? Since the differences between the proposed method and the others look minor for all three ablations, the readers can feel the environments can be cherry-picked. The reviewer thinks the authors need to show more results on different environments (can be in the appendix if space does not allow it).

**Questions:**

- Fig.2: The data distribution changes between stages 2 and 3 are a bit confusing. The left human to robot should happen in stage 2 and the right one should happen in stage 3, correct? If it is, the review thinks it would be clearer to enclose the left one with blue dots and the right one with green dots. Otherwise, it looks like the adaptations are independent of the proposed three stages and are confusing.
- Fig.2 and Sec.4: In stage 2, what do the dots between 'keypoint decoder' and 'reconstruct tgt view' mean? The reviewer read the implementation details shown in the appendix, but it does not explain the 'dots' part.
- The ratio of parameter changes is written as 0.36\% in the abstract and conclusion but written as 0.18\% in the other parts. Which is true or the reviewer misunderstands something?
- Fig.7: The reviewer cannot see the videos of how keypoints move in the video.
- Sec.5.2: The reviewer is curious about the performance of the BC-only policy while adapting the domain gap using expert trajectories. Does it correspond to the performance at environment steps being zero in Figure 4?

**Limitations:**

- As mentioned in the conclusion section, the method is tested on the same objects in the training set.

---

> ### Author Rebuttal · Authors · 2023-08-09
>
> We thank the reviewer for the constructive comments and suggestions. We address each of your comments in the following. We omit the questions due to space limits.
>
> **Q1:** L.145-146 .....
>
> **A1:** We have conducted a more comprehensive evaluation on our Stage 2 and the results are given in Figure 2 and Figure 4 of the rebuttal file. It is clearly shown that finetuning all parameters trivially could lead to suboptimal performance (Figure 2) and so is without any finetuning (Figure 4).
>
> **Q2:** L.209,212:.....
>
> **A2:** We would like to clarify that the statements mentioned might be misunderstood a bit by the reviewer, and we wanted to explain more here.
>
> ‘with appropriate domain knowledge, ConvNets can still outperform ViT’ With such statements, we are not trying to show ConvNets are better than ViT in general cases. *Instead, we want to show the advantage of our method*, that a feature representation with ConvNet architecture could beat other strong ViT-based methods including MVP and VC-1.
>
> ‘these observations highlight the task-dependent nature of the strengths exhibited by ConvNets and ViTs.’ This statement is saying that on different tasks, different architectures show very different results. For example, on *Relocate Tomato Soup Can*, RRL is much better than VC-1, while on *Relocate Mug*, it is the opposite. We agree with the reviewer that the statements are made not clear to fully express our observations, and would be more careful on the claims.
>
>
> **Q3:** L.273-276:.....
>
> **A3:** Previous studies in interpretability have shown that the deeper layers in neural networks have more high-level information. For example, Bau et al [2] show that color and texture concepts dominate at lower layers while more object and part detectors emerge in deeper layers; The visualizations in [3, 4] show the increase in complexity and variation on higher layers, while having simpler components from lower layers. In our experiments, we observe that the adaptation transforms the deep layer features more, thus we make the hypothesis that our method adapts the distribution of human hand features to the distribution of robot hand features, *especially in the higher-level parts*.
>
> **Q4:** Fig.7: ....
>
> **A4:** We apologize for the unclear explanation in the paper and give more explanations here. The term *keypoints* in our study refers to landmarks as defined in [1], which identify the most critical pixels for image reconstruction. As a result, some of these points might be located in unexpected positions. Additionally, the quantity of keypoints is preset (we use n=30 in our paper), implying the potential for redundant or idle points.
>
> **Q5:** Fig.6: .....
>
> **A5:** We acknowledge the insufficiency of the ablations and we want to respectfully emphasize that our ablation results are not cherry-picked; instead, the task is randomly sampled. During the rebuttal phase, we give a much more comprehensive evaluation of our Stage 2 and Stage 3. The results are given in the PDF file and clearly show the advantage of our method across tasks. We are also committed to releasing our codes.
>
> **Q6:** Fig.2: .....
>
> **A6:**  Thank the reviewer for the detailed suggestion on paper improvements. We have updated Figure 2 as suggested. The updated version is also given in the PDF file.
>
> **Q7:** Fig.2 and Sec.4: .....
>
> **A7:** We omit the complex self-supervised learning process in Figure 2 due to space limitation and explain briefly in Section 4.  Here we would like to give more intuitive explanations. More details could be found in [1].
>
> From a high perspective, the self-supervised learning task used in our paper is to reconstruct the target image, given a source image and a target image. The target image provides the keypoints, and the source image provides the appearance. For example, if we have a figure with a robot hand and the background (this is the source image), we could provide the new positions of the robot hand (encoded from the target image), and the information in the source image would be sufficient to reconstruct the target image where the robot hand is in the new position.
>
>
> **Q8:** The ratio of parameter changes....
>
> **A8:**  In Stage 2, the learnable parameters (weight, bias) in BatchNorm Layers are updated, which is 0.18% of the entire network. In Stage 3, moving mean and average parameters in BatchNorm are updated, which is also 0.18% of the entire network. These two parts together are 0.36%.
>
> **Q9:** Fig.7: The reviewer cannot see the videos of how keypoints move in the video.
>
> **A9:** The videos are displayed on our anonymous website during the initial submission. It could be correctly played in the Chrome browser of one MacBook or iPhone.
>
> **Q10:** Sec.5.2....
>
> **A10:** Yes. The performance of BC-only policy corresponds to the environment steps being zero.
>
> **Please do not hesitate to let us know if you have any additional comments.**
>
>
> [1] Jakab, Tomas, et al. "Unsupervised learning of object landmarks through conditional image generation." Advances in neural information processing systems 31 (2018).
>
> [2] Bau, David, et al. "Network dissection: Quantifying interpretability of deep visual representations." Proceedings of the IEEE conference on computer vision and pattern recognition. 2017.
>
> [3] Yosinski, Jason, et al. "Understanding neural networks through deep visualization." arXiv preprint arXiv:1506.06579 (2015).
>
> [4] Zeiler, Matthew D., and Rob Fergus. "Visualizing and understanding convolutional networks." Computer Vision–ECCV 2014: 13th European Conference, Zurich, Switzerland, September 6-12, 2014, Proceedings, Part I 13. Springer International Publishing, 2014.

---

> > ### Comment · Reviewer_rofX · 2023-08-14
> > **Reply by the reviewer**
> >
> > > QA1,2,3,6,7,8,9,10
> >
> > The reviewer wants to thank the authors for the clarifications.
> > Some other QAs still remain unclear.
> >
> > > QA4: "the quantity of keypoints is preset (we use n=30 in our paper), implying the potential for redundant or idle points."
> >
> > I think explicitly writing this in the paper would be better. Otherwise, readers (like me) will be confused.
> >
> > > QA5 " We acknowledge the insufficiency of the ablations and we want to respectfully emphasize that our ablation results are not cherry-picked; instead, the task is randomly sampled. During the rebuttal phase, we give a much more comprehensive evaluation of our Stage 2 and Stage 3. The results are given in the PDF file and clearly show the advantage of our method across tasks. We are also committed to releasing our codes."
> >
> > I don't think Fig.9 answers the original question while it still being very informative and comprehensive. Can the authors provide more results on ablations? Fig.9 is not related to Fig.6.

---

> > > ### Author Response · Authors · 2023-08-15
> > > **Thank you for the feedback!**
> > >
> > > We really appreciate your feedback! We would like to answer your questions as follows:
> > >
> > > ---
> > >
> > > **QA4:** "the quantity of keypoints is preset (we use n=30 in our paper), implying the potential for redundant or idle points."
> > >
> > > I think explicitly writing this in the paper would be better. Otherwise, readers (like me) will be confused.
> > >
> > > **A:** Yes, we agree with the reviewer and we would add this detail about Stage 2 in our revised paper. Thank you for the suggestion.
> > >
> > > ---
> > >
> > > **QA5:** “We acknowledge the insufficiency of the ablations and we want to respectfully emphasize that our ablation results are not cherry-picked; instead, the task is randomly sampled. During the rebuttal phase, we give a much more comprehensive evaluation of our Stage 2 and Stage 3. The results are given in the PDF file and clearly show the advantage of our method across tasks. We are also committed to releasing our codes."
> > >
> > > I don't think Fig.9 answers the original question while it still being very informative and comprehensive. Can the authors provide more results on ablations? Fig.9 is not related to Fig.6.
> > >
> > > **A:** Yes, Figure 9 (found in the supplementary file) is not related to the reviewer’s question, and it was also not our intention. This figure (fig. 9) was presented in our initial submission. For the rebuttal phase, we conducted extensively more ablation studies, and results could be found in the attachment under **Author Rebuttal by Authors**. We kindly ask the reviewer to consider our updated results, which comprehensively show the advantages of our proposed Stage 2 and Stage 3.
> > >
> > > ---
> > >
> > > **As the rebuttal phase draws to a close, we sincerely hope our responses have addressed the reviewer's concerns. We kindly hope the reviewer could raise the score if the concerns are addressed. If the reviewer has any additional comments, please do not hesitate to let us know.**

---

> > > > ### Comment · Reviewer_rofX · 2023-08-15
> > > > **Thanks for the further clarification.**
> > > >
> > > > > A: Yes, Figure 9 (found in the supplementary file) is not related to the reviewer’s question, and it was also not our intention. This figure (fig. 9) was presented in our initial submission. For the rebuttal phase, we conducted extensively more ablation studies, and results could be found in the attachment under Author Rebuttal by Authors. We kindly ask the reviewer to consider our updated results, which comprehensively show the advantages of our proposed Stage 2 and Stage 3.
> > > >
> > > > Oh, I'm sorry I checked the wrong file. Yes, the attachment would be great, properly reflecting my comments and answering my questions. Now I would like to increase my rates.
> > > >
> > > > Thanks again for the constructive discussion.

---

> > > > > ### Author Response · Authors · 2023-08-15
> > > > > **We appreciate your quick and positive feedback!**
> > > > >
> > > > > We are happy that our replies address your questions!
> > > > >
> > > > > We sincerely thank the reviewer for the kind response and the positive feedback.

---

### Official Review · Reviewer_s3to · 2023-07-05

**Soundness:** 3 good
**Presentation:** 3 good
**Contribution:** 1 poor
**Rating:** 4
**Confidence:** 4

**Summary:**

This paper studies a three stage framework for adapting pre-trained visual representations for dexterous manipulation tasks. Specifically, this work proposes only adapting a subset of network parameters in stages 2 and 3. The approach is tested on 12 manipulation tasks from Adroit and DexMV. These experiments demonstrate that the proposed techniques lead to more sample efficient learning. Furthermore, the overall approach outperforms alternative pre-training methods from recent work.

**Strengths:**

- This paper demonstrates that using a visual backbone pretrained for (human) hand-pose estimation leads to improvements in learning downstream dexterous manipulation tasks over strong baselines from recent work. These results suggest that this inductive bias might be missing from recently proposed methods.
- This work proposes adapting a specific subset of parameters during stages 2 and 3 of training, and demonstrates that this leads to more efficient learning.
- The paper is clearly written and most of the details of the proposed approach are clear.

**Weaknesses:**

Large parts of the proposed approach are directly applied from prior work including the pretrained visual representation (stage 1) and the learning objective from stage 2. This is clearly stated in the manuscript and appendix, and is not in itself an issue. However, this means the contributions of this work are primarily in the parameter efficient techniques used in stage 2 and 3.

- Unfortunately, the results in Figure 6 indicate that stage 2 may improve sample efficiency, but may not improve asymptotic performance. Thus, it is unclear if this is a significant contribution.
- Furthermore, the sample efficiency gains (Figure 6) are only demonstrated on two tasks (Hammer and Place Inside), so it is unclear how general these findings are.
- Similarly, when compared to full finetuning (Figure 6c) it is not clear if asymptotic performance is improved and the sample efficiency gains are minimal.

**Questions:**

- How does asymptotic performance compare in the ablations in Figure 6?
- Do the sample efficiency improvements generalize to other tasks?

**Limitations:**

Yes, the authors discuss limitations of the work.

---

> ### Author Rebuttal · Authors · 2023-08-09
>
> We thank the reviewer for the constructive comments and suggestions. We address each of your comments in the following.
>
> **Q1:** Large parts of the proposed approach are directly applied from prior work including the pre-trained visual representation (stage 1) and the learning objective from stage 2. This is clearly stated in the manuscript and appendix, and is not in itself an issue. However, this means the contributions of this work are primarily in the parameter efficient techniques used in stage 2 and 3.
>
> **A1:** We thank the reviewer for insightful comments and for acknowledging our contributions in stages 2 and 3 of the proposed approach. We greatly value your feedback.
>
> However, we would like to respectfully point out that Stage 1, the usage of a pre-trained representation using 3D human hand pose estimation is also one of our main contributions. While it is true that Stage 1 simply employs an existing model, **the inductive bias, introduced by us is not trivial and still intuitive**. A lot of recent works stick to using unsupervised learning on egocentric data to pretrain a new task-agnostic model, such as MVP [1,4], R3M [2], VIP [3], and RPT  [5]. However, we show that with the proper usage, the existing pre-trained model from other domains could be enough. **Moreover, it's not straightforward to adapt a pre-trained model from different domains.** As shown in Figure 5 of the paper, some other models that also share a similar intuition, such as AlphaPose, could not gain a reasonable result. And as shown in our new ablation results (Figure 2, Figure 3, and Figure 4 in the rebuttal file), wrongly using the pre-trained model would not lead to reasonable performance. **Our framework, combining all three stages together and correctly adapting the pre-trained model, unleashes the potential of the pre-trained encoder.**
>
> We believe our Stage 1 and the efforts in combining all three stages effectively are also the core contributions of our work.
>
> [1] Xiao, Tete, et al. "Masked visual pre-training for motor control." arXiv preprint arXiv:2203.06173 (2022).
>
> [2] Nair, Suraj, et al. "R3m: A universal visual representation for robot manipulation." arXiv preprint arXiv:2203.12601 (2022).
>
> [3] Ma, Yecheng Jason, et al. "Vip: Towards universal visual reward and representation via value-implicit pre-training." arXiv preprint arXiv:2210.00030 (2022).
>
> [4] Radosavovic, Ilija, et al. "Real-world robot learning with masked visual pre-training." Conference on Robot Learning. PMLR, 2023.
>
> [5] Radosavovic, Ilija, et al. "Robot Learning with Sensorimotor Pre-training." arXiv preprint arXiv:2306.10007 (2023).
>
>
> **Q2:** Unfortunately, the results in Figure 6 indicate that stage 2 may improve sample efficiency, but may not improve asymptotic performance. Thus, it is unclear if this is a significant contribution. Furthermore, the sample efficiency gains (Figure 6) are only demonstrated on two tasks (Hammer and Place Inside), so it is unclear how general these findings are. Similarly, when compared to full finetuning (Figure 6c) it is not clear if asymptotic performance is improved and the sample efficiency gains are minimal.
> How does asymptotic performance compare in the ablations in Figure 6? Do the sample efficiency improvements generalize to other tasks?
>
> **A2:** During the rebuttal phase, we have conducted a more comprehensive evaluation of our proposed Stage 2 and Stage 3, across almost all tasks, as shown in our rebuttal PDF file (Figure 2, Figure 3, and Figure 4).
>
> In Figure 2, we conduct experiments across all 12 tasks and show that adapting all parameters in Stage 2 is not better than our proposed method in Stage 2, which only adapts 0.18% parameters. It is also observed that on Relocate Box, adapting all parameters is better, but this phenomenon only appears in one task.
>
> In Figure 3, we conduct experiments across all 7 tasks that use m>0 in Stage 3. It is shown that our adaption in Stage 3 also makes a solid contribution to the final results achieved.
>
> In Figure 4, we show that directly removing Stage 2 instead of full finetuning (which is shown in Figure 2) is also not good. This observation is natural since we use in-domain data to finetune the pre-trained model, while it could be generally shown that wrongly finetuning the model (Figure 2) could possibly hurt more than not finetuning the model (Figure 4).
>
> In response to the reviewer's comment on asymptotic performance, we present the convergence results in Figure 7. This indicates that our method reaches the peak performance at 2M steps, so we saw no need to continue further steps.
>
> Overall, we enhance the ablations during the rebuttal phase and we hope that these new results could show the advantages of our proposed techniques in improving the pre-trained vision model for downstream RL tasks.
>
> **Please do not hesitate to let us know if you have any additional comments.**

---

> ### Author Response · Authors · 2023-08-20
> **Thank you for the review and awaiting your response**
>
> We sincerely thank you for your efforts in reviewing our paper and the suggestions again.
>
> We believe that we have resolved all the concerns mentioned in the review. Should there be any additional concerns, we are more than happy to address them! Thank you very much!

---

> > ### Comment · Reviewer_s3to · 2023-08-21
> > **Thanks for the response**
> >
> > Thank you for the response. Most of my questions are resolved, but I am still somewhat concerned about asymptotic performance. I will update my score accordingly.
> >
> > In Figure 7 in the rebuttal pdf it appears that with enough environment steps, "stage 1" and "stage 1+2+3" may converge. This would of course mean that stages 2 and 3 are not needed. That said, it seems likely that the "stage 1" curve will saturate at or below the "stage 1+2" curve. While this remains an open, empirical question, these additional ablations are encouraging.

---

> > > ### Author Response · Authors · 2023-08-21
> > > **Thank you for the feedback! We address your questions as below.**
> > >
> > > We thank the reviewer for the feedback. We agree with the reviewer that it could be good to see also results of more steps for *Stage 1 only*. During the rebuttal phase, we also run *Stage 1 only* for longer steps.  We provide an updated version of Figure 7 ([link](https://drive.google.com/file/d/1erAasSzVJOowc2z8HVvBwaeOJ_E49OXo/view?usp=sharing)). It is obviously observed that our method achieves the highest score with only 2M steps, while *Stage 1 only* gets similar or slightly worse results with 6M steps.
> > >
> > > Also, it should be acknowledged that on Hammer, *Stage 1 only* has been strong enough, which is exactly our original motivation to explore this pre-trained representation.
> > >
> > >
> > > Additionally, we kindly hope the reviewer could also consider our new ablation results in Figure 2, Figure 3, and Figure 4 of the rebuttal file. We show that on various tasks, removing Stage 2 or Stage 3 will lead to sub-optimal performance. Consequently, Stage 2 and Stage 3 are not only necessary but also crucial for our final performance, instead of "not needed".
> > >
> > > We would update all these results in our future revision. **We believe all our contributions are solid and demonstrated across diverse challenging dexterous manipulation tasks**.  If the reviewer has any additional comments, please do not hesitate to let us know and we are happy and eager to address them.

---

### Official Review · Reviewer_GX9c · 2023-07-07

**Soundness:** 3 good
**Presentation:** 3 good
**Contribution:** 2 fair
**Rating:** 6
**Confidence:** 4

**Summary:**

This paper introduces a framework called H-InDex that uses human hand-inspired visual representation learning to solve complex manipulation tasks with reinforcement learning. The framework consists of three stages and outperforms other methods in challenging dexterous manipulation tasks.

**Strengths:**

* A novel and effective method to perform dexterous hand manipulation using visual RL.

* Extensive experiments and ablation study on various tasks.

* A convincing framework exploiting the pre-trained vision models for robot learning.

**Weaknesses:**

* The method seems only to be able to perform on a single object on each task, while the vision model should be able to generalize on category-level objects.

* It is unclear of the RL action space, reward design, robotic control parameters, and the coordinate system used in the paper.

**Questions:**

* Are the proprioceptive states of the robotic hand (e.g. joint states) considered in stage 3?

* How does this work encode the object information in the compact representations?

**Limitations:**

The authors have discussed the limitations.

---

> ### Author Rebuttal · Authors · 2023-08-09
>
> We thank the reviewer for the constructive comments and suggestions. We address each of your comments in the following.
>
> **Q1:** The method seems only to be able to perform on a single object on each task, while the vision model should be able to generalize on category-level objects.
>
> **A1:** Achieving generalization across objects is not trivial, especially when the training objects are fixed and not diverse. In this work, all our tasks only have one single object and we do not put our focus on generalizable manipulation (clearly stated in the limitation part). Instead of generalizable manipulation, one good property of the pre-trained visual presentation is visual generalization, such as generalizing to unseen scenes. To show our model’s visual generalization ability, we change the visual backgrounds and evaluate our H-InDex and VC-1 on Relocate Potted Meat Can task. The results are given in Table 2 of the rebuttal file and the visualizations are also given in Figure 6. We could observe that, though VC-1 has a slightly better training performance on this task, our method presents better robustness on these new scenes, indicating the strong visual generalization ability of H-InDex.
>
> **Q2:** It is unclear of the RL action space, reward design, robotic control parameters, and the coordinate system used in the paper.
>
> **A2:** In this work, all our tasks follow prior works (Adroit [1] and DexMV [2]) and we do not make specific changes to the tasks, such as action space and reward design. The action space is 30-dim and the action controls the joints of a five-finger robot hand, i.e., Adroit hand. We use a dense reward for RL and the reward design is task-specific. We will add more task details in our revised version.
>
> **Q3:** Are the proprioceptive states of the robotic hand (e.g. joint states) considered in stage 3?
>
> **A3:** Yes, and the robot hand states are only used in Stage 3 (RL stage). In our paper, we also mention the usage of robot hand states in Section 3 (L102). In addition, the usage of robot hand states is a common practice in visual-based robot learning [3, 4, 5], and all our baseline methods also use robot states for fair comparison.
>
> **Q4:** How does this work encode the object information in the compact representations?
>
> **A4:** The object information is directly encoded from the visual observations. We do not use any other information such as the object bounding box, the object coordinates, and so on. This makes the learning process extremely challenging, especially for dexterous manipulation.
>
> **Please do not hesitate to let us know if you have any additional comments.**
>
> [1] Rajeswaran, Aravind, et al. "Learning complex dexterous manipulation with deep reinforcement learning and demonstrations." arXiv preprint arXiv:1709.10087 (2017).
>
> [2] Qin, Yuzhe, et al. "Dexmv: Imitation learning for dexterous manipulation from human videos." European Conference on Computer Vision. Cham: Springer Nature Switzerland, 2022.
>
> [3] Shah, Rutav, and Vikash Kumar. "Rrl: Resnet as representation for reinforcement learning." arXiv preprint arXiv:2107.03380 (2021).
>
> [4] Xiao, Tete, et al. "Masked visual pre-training for motor control." arXiv preprint arXiv:2203.06173 (2022).
>
> [5] Radosavovic, Ilija, et al. "Real-world robot learning with masked visual pre-training." Conference on Robot Learning. PMLR, 2023.

---

> > ### Comment · Reviewer_GX9c · 2023-08-20
> >
> > I appreciate the author's response and will maintain my ratings.

---

> > > ### Author Response · Authors · 2023-08-21
> > > **Thank you for maintaining the score as weak accept!**
> > >
> > > We sincerely appreciate your feedback and understand from your comments that our rating stands at *Weak Accept*.
> > >
> > > However, we observed that the score in the review has been adjusted to *Borderline Accept*. We wonder if this might have been an unintended change. If that's the case, could the reviewer kindly revert the score?
> > >
> > > We have diligently addressed all the concerns previously mentioned. If there are any further issues or feedback, we are happy and eager to address them promptly.

---

### Author Rebuttal · Authors · 2023-08-09

We thank all the reviewers for their insightful comments. We have addressed all your individual comments. We wanted to thank the reviewer for acknowledging the novelty and the empirical evaluation of our work – “a novel and effective method” (GX9c), “experiments are extensively conducted” (rofX), “interesting and well-motivated” (di2q), “improvements over strong baselines” (s3to). More additional experiments are also conducted during the rebuttal phase to support our proposed method (given in the PDF file), as suggested by the reviewers.

**EXP1: Ablation on Stage 2 (adapting 0.18% parameters or adapting 100% parameters)** in reply to Reviewer s3to and Reviewer rofX. Results are given in Figure 2. We compare *only adapting 0.18% parameters in Stage 2* (our proposed method) with *adapting all parameters*, across all 12 tasks. It is clearly shown that finetuning all parameters could lead to suboptimal performance and even make agents fail to learn on *Door, Relocate Large Clamp, and Relocate Potted Meat Can*. We also observe that on Relocate Box task, finetuning 100% parameters is surprisingly more effective, but considering the high variance and the low average scores of finetuning 100% parameters across 12 tasks, the advantage of our proposed adaptation is clearer.

**EXP2: Ablation on Stage 3 (m > 0 or m = 0)** in reply to Reviewer s3to and Reviewer rofX, Reviewer di2q. Results are given in Figure 3. We compare setting *m > 0* with *m=0* across all 7 tasks where m is set to be larger than 0. It is observed that adding this simple technique (updating statistics with momentum) during the RL phase could have a large impact. The empirical results emphasize the importance of correct adaptation when applying the pre-trained model. We do not conduct more ablations on the rest 5 tasks that use m=0 originally, since we currently do not find a suitable m\in {0.1, 0.01, 0.001} that could outperform m=0, which means the visual representation without adaption in the RL phase has been good enough.

**EXP3: Ablation on Stage 2 (with or without adaptation)** in reply to Reviewer s3to and Reviewer rofX. In addition to experiments on adapting 100% parameters, we also show more results about not adding Stage 2 in Figure 4. The experiments cover all 6 kinds of tasks in our paper, and we think it is extensive to show the necessity of our Stage 2. It is observed that except Door task (where results are similar), our method with Stage 2 outperforms that without Stage 2 across all other tasks.

**EXP4: Visual Generalization to unseen backgrounds** in reply to Reviewer GX9c. Results are given in Table 2 and visualizations are given in Figure 6. The experiments are conducted on Relocate Potted Meat Can task, where the best baseline VC-1 is slightly better than our H-InDex. However, we show that when the backgrounds change from seen to unseen, VC-1 is distracted more than H-InDex, and H-InDex achieves 475.04 average scores across 9 new scenes while VC-1 has only 392.56 scores. This experiment shows that even after adaptation, our visual representation does not overfit to training data simply and still keeps good generalization ability.

Again, we thank the reviewers for their constructive feedback. We believe that all individual comments have been addressed, but are happy to address any further comments from reviewers.

---

### Decision · Program_Chairs · 2023-09-21

**Decision:**

Accept (poster)

**Comment:**

The paper proposes a three-stage framework for dexterous manipulation using human hand-inspired visual representation learning. The initial concerns from the reviewers include: 1) unclear details of the method and statements, 2) unclear contributions on the three stages, 3) missing ablations on multiple tasks, and 4) marginal improvements of stages 2&3 and inferior performance on certain tasks.
The rebuttal addressed most of these concerns by clarifying the method and providing additional results. In the end, there are two weak accept, 1 borderline accept and 1 borderline reject. The main concern from the two reviewers who gave borderline scores is the suboptimal performance.
The AC considers the proposed method provides some new insights on pretrained representations for dexterous manipulation and the experiments support the claims.
Therefore, the AC recommends accepting the paper.
The authors should include the new results and discussions according to reviewers' comments in their final version.